# Comparative Transcriptome Analysis Reveals the Protective Role of Melatonin during Salt Stress by Regulating the Photosynthesis and Ascorbic Acid Metabolism Pathways in *Brassica campestris*

**DOI:** 10.3390/ijms25105092

**Published:** 2024-05-07

**Authors:** Sayyed Hamad Ahmad Shah, Haibin Wang, Huanhuan Xu, Zhanghong Yu, Xilin Hou, Ying Li

**Affiliations:** 1National Key Laboratory of Crop Genetics and Germplasm Enhancement and Utilization, Nanjing Agricultural University, Nanjing 210095, China; 2018204050@njau.edu.cn (S.H.A.S.); 2020204029@stu.njau.edu.cn (H.W.); 2022204057@stu.njau.edu.cn (H.X.); 2018204024@njau.edu.cn (Z.Y.); hxl@njau.edu.cn (X.H.); 2Engineering Research Center of Germplasm Enhancement and Utilization of Horticultural Crops, Ministry of Education, Nanjing 210095, China; 3College of Horticulture, Nanjing Agricultural University, Nanjing 210095, China

**Keywords:** *Brassica campestris*, melatonin treatment, salt stress, biochemical changes, transcriptome analysis

## Abstract

Salinity stress is a type of abiotic stress which negatively affects the signaling pathways and cellular compartments of plants. Melatonin (MT) has been found to be a bioactive compound that can mitigate these adverse effects, which makes it necessary to understand the function of MT and its role in salt stress. During this study, plants were treated exogenously with 100 µM of MT for 7 days and subjected to 200 mM of salt stress, and samples were collected after 1 and 7 days for different indicators and transcriptome analysis. The results showed that salt reduced chlorophyll contents and damaged the chloroplast structure, which was confirmed by the downregulation of key genes involved in the photosynthesis pathway after transcriptome analysis and qRT-PCR confirmation. Meanwhile, MT increased the chlorophyll contents, reduced the electrolyte leakage, and protected the chloroplast structure during salt stress by upregulating several photosynthesis pathway genes. MT also decreased the H_2_O_2_ level and increased the ascorbic acid contents and APX activity by upregulating genes involved in the ascorbic acid pathway during salt stress, as confirmed by the transcriptome and qRT-PCR analyses. Transcriptome profiling also showed that 321 and 441 DEGs were expressed after 1 and 7 days of treatment, respectively. The KEGG enrichment analysis showed that 76 DEGs were involved in the photosynthesis pathway, while 35 DEGs were involved in the ascorbic acid metabolism pathway, respectively. These results suggest that the exogenous application of MT in plants provides important insight into understanding MT-induced stress-responsive mechanisms and protecting *Brassica campestris* against salt stress by regulating the photosynthesis and ascorbic acid pathway genes.

## 1. Introduction

Natural plant environments are surrounded by many abiotic and biotic stresses regularly, which affect normal plant growth and development. Salinity stress is a type of abiotic stress that negatively affects the signaling pathways and cellular compartments of plants [1,2]. According to previous estimations, 6% of the total world area and 20% of irrigated land have been adversely affected due to high salinity [3]. High salt concentrations can cause osmotic stress, ion toxicity, nutrient imbalance, and oxidative damage in plants, which can lead to reduced photosynthetic efficiency and impaired plant growth [4,5]. Chlorophyll a (chl-a) and chlorophyll b (chl-b) pigments play a key role in the absorption and transmittance of light energy during the photosynthesis process [6]. Chl-a and chl-b molecules are bound by CAB proteins, which particularly increase light absorption in the blue and red portions of the spectrum and are essential for the photosynthesis process [7]. Thus, any interference with chloroplast metabolism and function impairs the effectiveness of light-induced stimulation energy transfer and, in turn, decreases the rate of CO_2_ assimilation [8]. Salt stress has direct impacts on photosynthesis, such as diffusion restrictions via the stomata and mesophyll cells, which change the photosynthetic metabolism [8,9]. High levels of Na^+^ uptake by the cytoplasm decrease metabolic and various enzymes’ activities and also inhibit the photosynthesis process at the cellular level [9,10]. The effects of salinity on photosynthesis may include a reduction in electron transfer, the inactivation of the photosystem II reaction centers, the destruction of the oxygen-evolving complex, and a reduction in the donor side’s ability to transfer electrons [11,12].

Melatonin (*N*-acetyl-5 methoxytryptamine) enhances photosynthetic ability and protects the plant photosystem by inhibiting chlorophyll degradation, upregulating chlorophyll-synthesis-related genes, and maintaining the fluidity of the thylakoid membrane [13]. A variety of plant species have been found to synthesize melatonin (MT) and are used for different activities [13,14]. MT has been found in the roots, leaves, fruits, and seeds of several plant species, including the Brassicaceae family. Researchers have more interest in MT due to its several biological functions, like as a plant signaling molecule and its protective role in unpredictably stressful conditions [15]. Additionally, MT speeds up seed germination [16], has an impact on root and plant architecture, improves growth vigor, reduces leaf senescence, and adjusts physiological activities through regulating various genes [17,18]. Studies have revealed many functions of MT, such as the photoperiodic response and regulation of plant reproductive physiology, the defense of plant cells against apoptosis caused by severe environments, and the ability to participate as a free radical scavenger and/or upregulator of certain defensive enzymes in the senescent process [19,20]. MT has been found to alleviate salt stress in plants by reducing oxidative stress and increasing the activity of antioxidant enzymes [21,22]. MT application causes a modest rise in endogenous indole-3-acetic acid levels when compared to untreated plants, as seen in *Brassica juncea* [23] and tomato plants [24]. MT, as a free radical scavenger and antioxidant, plays critical functions in improving antioxidant systems under salt stress, i.e., enhancing photosynthesis, ion homeostasis, and antioxidant enzymes, activating a variety of downstream hormones and polyamine metabolism, and regulating the expression of stress-responsive genes [15,23,25].

*Brassica campestris* (*B. campestris*) is an important leafy vegetable that is widely cultivated in many parts of Asia [26,27]. However, its growth and productivity are often limited by various environmental stresses, such as salinity, drought, and high temperature, causing physiological, molecular, and biochemical changes in plant growth and yield [28,29]. MT has been found to be a potent bioactive compound recently and plays a protective role in salt stress by enhancing antioxidant substances in plants. In this regard, we conducted this study to identify the protective role of melatonin during salt stress in *B. campestris* through biochemical and transcriptome analyses to provide new insights for future studies.

## 2. Results

### 2.1. Exogenous Melatonin (MT) Application Reduced Electrolyte Leakage and Protected the Photosynthetic Pigments during Salt Stress

To investigate the protective effects of melatonin (MT) against salt stress in *Brassica campestris* (*B. campestris*), we analyzed electrolyte leakage, chl-a, chl-b, and the total chlorophyll content. The findings revealed that electrolyte leakage, chl-a, chl-b, and the total chlorophyll content were significantly affected by both MT and salt stress as compared to the control (Figure 1). While salt stress led to increased electrolyte leakage and decreased chlorophyll contents across all the treatments, the negative effects were notably mitigated by the application of exogenous MT. This suggests that the exogenously applied MT during salt stress protected the photosynthetic pigments in *B. campestris*.

### 2.2. Exogenous MT Protected the Chloroplast Structure of B. campestris during Salt Stress

Exposure to salt stress led to notable damage to chloroplasts, the membrane structure, and stroma lamellae, which are the main components of the plant photosystem (Figure 2). However, the application of MT alongside salt strengthened the plant chloroplasts, the membrane structure, and stroma lamellae, indicating the effectiveness of MT in alleviating salt stress. Our findings suggest that the plants treated with MT showed a natural chloroplast structure and stroma lamellae that control plants. In contrast, under salt stress conditions, the chloroplast structure and stroma lamellae were extensively damaged, whereas the plants treated with the foliar application of MT during salt stress preserved their integrity and structure in these cellular components. This observation suggests that MT plays a protective role in maintaining cell integrity and chloroplast structure amidst salt stress.

### 2.3. Exogenous MT Application Enhanced the Ascorbic Acid Content and APX Activity and Reduced the ROS Level during Salt Stress

To further investigate the protective effect of exogenous MT application in salt stress, we analyzed the accumulation of ascorbic acid content, the APX activity, and the ROS level. Our findings demonstrated that salt stress with MT in comparison to the control led to notable increases in AsA, total AsA, DHA, and APX activity and decreased the ROS level in *B. campestris* (Figure 2). Compared to the control, the plants that were subjected to salt stress in combination with MT treatment showed higher ascorbic acid contents and APX activity after 1 and 7 days, suggesting that MT application increased the plants’ stress tolerance by enhancing the antioxidant enzymes’ activities during salt stress. MT also reduced the ROS level during salt stress. Compared to the control, the H_2_O_2_ content and DAB staining were lower in the plants treated with MT after 7 days. Subsequently, compared to the control group, the plants exposed to salt exhibited a higher H_2_O_2_ content and deeper staining after 7 days. In comparison to the control plants, the MT-co-treated plants with salt showed deeper staining and higher H_2_O_2_ content, but less than the salt-treated plants after 7 days of treatment, suggesting that MT reduced the ROS level to minimize salt stress (Figure 2).

### 2.4. RNA-Seq and DEG Identification

A total of 24 cDNA libraries were constructed from the CK (control), MT (melatonin), ST (NaCl), and SMT (NaCl + melatonin) plant groups. In total, 44.64 million reads and 6695.73 trillion bases were found, of which 44.1717 million were clean reads and 6545.54 million were clean bases (Appendix A). For the DEG analysis, a pairwise comparison was used to evaluate the up- and downregulation of DEGs, as shown in Figure 3. The number of DEGs was highly significant after the melatonin and salt treatment at different intervals. The volcano diagram for CK1_vs_MT1 and CK7_vs_MT7, CK1_vs_ST1 and CK7_vs_ST7, CK1_vs_SMT1 and CK7_vs_SMT7 showed that most of the genes were downregulated. In the volcano diagram, the red color shows the upregulation, and the green color shows the downregulation of DEGs. Next, we identified the differential expression of genes in the CK_vs_MT, ST, and SMT groups at different intervals to find the total number of DEGs expressed, as shown in Figure 4. A total of 321 DEGs for CK1_vs_MT1, CK1_vs_ST1, and CK1_vs_SMT1 were significantly expressed after 1 day of treatment. Furthermore, 441 DEGs were found for CK7_vs_MT7, CK7_vs_ST7, and CK7_vs_SMT7 after 7 days of treatment, which was higher than after day 1, suggesting that after 7 days of MT, the NaCl and NaCl + MT treatments triggered the expression of a greater number of DEGs. 

### 2.5. Gene Ontology Enrichment Analysis of the DEGs

For the gene ontology enrichment analysis, a pairwise analysis was carried out for the CK (control) group with all the treatments after 1 and 7 days, as shown in Figure 5. The DEGs were divided into biological processes, cellular components, and molecular functions based on the GO enrichment analysis. In the CK1_vs_MT1 group, the top four enriched biological processes were cellular processes, metabolic processes, single-organism processes, and responses to stimuli, and these were found to be downregulated. In terms of cellular components, the top four enrichments were cells, cell parts, organelles, and membranes, and these were found to be downregulated. In terms of molecular function, the top four enrichments involved binding, catalytic activity, transporter activity, and nucleic acid binding transcription factor activity; whereas transporter activity genes were found to be upregulated, the others were downregulated after 1 day of treatment. In the CK7_vs_MT7 group, the top four enrichments involved in biological processes were found to be the same as for the CK1_vs_MT1 group, but the genes for cellular and metabolic processes were found to be upregulated. In terms of cellular components, the top four enrichments were cells, cell parts, organelles, and organelle parts, where the organelle part genes were found to be upregulated and the other components’ genes were found to be downregulated. In terms of molecular functions, the top four enrichments involved binding, catalytic activity, structural molecule activity, and nucleic acid binding transcription factor activity, where all genes were found downregulated, except for those involved in structural molecule activity. In the CK1_vs_ST1 group, the same components were found for biological processes, cellular components, and molecular functions as in the CK7_vs_ST7 group, and the downregulation ratio for biological process, cellular component, and molecular function genes was found to be higher, but not that for transporter activity genes, which were upregulated in the CK7_vs_ST7 group. In the CK1_vs_SMT1 and CK7_vs_SMT7 groups, similar components were identified in biological processes, cellular components, and molecular functions, and most genes were found to be downregulated, with the exception of transporter activity genes, showing that most genes in all the comparative groups were downregulated in our experiment.

### 2.6. KEGG Enrichment Analysis of DEGs

Based on the KEGG enrichment analysis, DEGs in *B. campestris* leaves were involved in biological, metabolic, and signal transduction pathways under the MT and salt treatments (Figure 6). It was found that the highest number of genes was found in the metabolic and biosynthesis pathways of secondary metabolites in the CK1_vs_MT1, CK1_vs_ST1, CK7_vs_ST7, CK1_vs_SMT1, and CK7_vs_SMT7 groups. In the CK7_vs_MT7 comparison group, the maximum number of genes was identified in the ribosome pathway, while on the basis of the enrichment ratio, glucosinolate biosynthesis was a significantly more enriched pathway in the same group. However, stilbenoid, diarylheptanoid, and gingerol biosynthesis, porphyrin metabolism, and photosynthesis-antenna protein pathways were significantly enriched in the CK1_vs_ST1 group, while styrene degradation, photosynthesis-antenna proteins, and photosynthesis were the most significantly enriched pathways in the CK7_vs_ST7 group. Moreover, photosynthesis-antenna proteins and O-antigen nucleotide sugar metabolism were significantly enriched in the CK1_vs_SMT1 group, while styrene degradation, phenylalanine metabolism, glyoxylate and dicarboxylate metabolism, carbon fixation in photosynthetic organisms, and alpha-linolenic acid metabolism pathways were found to be significantly enriched in the CK7_vs_SMT7 group.

### 2.7. Photosynthesis-Related Gene Network Analysis of B. campestris Leaves after MT Application during Salt Stress

Photosynthesis is associated with four types of membrane protein complexes: ATPase, cytochrome b6f, photosystem II (PSII), and photosystem I (PSI), as shown in Figure 7. The differential expression of 27, 19, 18, and 12 genes in PSII, PSI, photosynthetic electron transport, and F-type ATPase was shown by the KEGG enrichment analysis in all the comparison groups in our study. In PSII, all 27 genes were upregulated in all the comparison groups. However, in the CK7_vs_MT7 group, 16 genes were upregulated, while 11 genes were downregulated. In the CK7_vs_ST7 group, one gene (*BraC10g005390*) was upregulated, and all other genes were downregulated, and the same trend was followed by the CK7_vs_SMT7 group, where only one gene (*BraC02g018500*) was upregulated, which means that these genes have a crucial role in the PSII under stress conditions. In PSI, out of 19 genes, 1 gene (*BraC10g008090*) was downregulated in the CK1_vs_MT1 group, while in the other groups, all the genes were upregulated. However, in the CK7_vs_MT7 group, 10 genes were upregulated, while 9 genes were downregulated in PSII. In the CK7_vs_ST7 group, two genes (*BraC10g008090* and *BraC04g009770*) were upregulated, while in the CK7_vs_SMT7 group, one of the same genes (*BraC10g008090*) was upregulated as in the CK7_vs_ST7 group, which means that these genes have a very vital role in PSI during salt stress. In terms of photosynthetic electron transport genes, three genes (*BraC03g025430, BraC04g021990*, and *BraC07g018060*) were downregulated in the CK1_vs_MT1 and CK1_vs_ST1 groups, while in the CK1_vs_SMT1 group, four genes (*BraC03g025430*, *BraC04g021990*, *BraC07g018060*, and *BraC09g037420*) were downregulated. However, in the CK7_vs_MT7 group, 11 genes were upregulated, while 7 genes were found to be downregulated. In terms of F-type ATpase genes, three genes (*BraC01g042040*, *BraC08g030620*, and *BraC07g014400*) were downregulated in the CK1_vs_MT1 group, two genes (*BraC05g022800* and *BraC07g014400*) in the CK1_vs_ST1 group, and one gene (*BraC05g022800*) in the CK1_vs_SMT1 group. However, in the CK7_vs_MT7 group, nine genes were found to be upregulated. In the CK7_vs_ST7 group, five genes were upregulated, while in the CK7_vs_SMT7 group, six genes were upregulated, which demonstrates that these genes had a very important role in *B. campestris* during salt stress when MT was applied.

### 2.8. Ascorbic Acid Metabolism Network in B. campestris Leaves after MT Application during Salt Stress

We examined the expression pattern of associated DEGs in order to understand more about the impact of MT and salt stress on ascorbic acid metabolism at the transcriptional level, as shown in Figure 8. It was found that 35 DEGs were involved in the ascorbic acid metabolism pathway. In the CK1_vs_MT1 comparison group, 23 genes were upregulated, while 12 genes were downregulated. However, in the CK7_vs_MT7 group, 13 genes were upregulated, and 22 genes were downregulated. In the CK1_vs_ST1 group, 19 genes were upregulated, while 16 genes were downregulated. In the CK1_vs_SMT1 group, 19 genes were upregulated, and the remaining 16 genes were downregulated. Moreover, in the CK7_vs_SMT7 group, 22 genes were upregulated, which was higher than in the CK1_vs_SMT1 group, demonstrating that these upregulated genes may have a crucial role in ascorbic acid metabolism under salt stress when MT is applied.

### 2.9. Quantitative Real-Time PCR Analysis

To verify the validity and reliability of the genes identified from our RNA-seq data, several DEGs related to photosynthesis and the ascorbic acid metabolism pathway were selected, and their expression levels were investigated in the different experimental groups under both melatonin and salt stress conditions using qRT-PCR analysis. As shown in Figure 9, the relative expression level of the selected genes showed a consistent pattern with the RNA-seq analysis. For instance, the genes *BraC02g004120*, *BraC02g042580*, and *BraC02g018500* showed upregulation in the MT-co-treated plants with salt after 1 day of treatment, and they also showed upregulation in the RNA-seq data. Similarly, the genes *BraC10g008090* and *BraC03g059550* showed downregulation, as also depicted in the RNA-seq data, when compared with the plants treated with only salt. From these findings, we can infer that the quality, accuracy, and reliability of the transcriptome data are adequate for designing future functional studies.

## 3. Discussion 

The synthesis of enzymes involved in photosynthesis and chlorophyll depends on various environmental conditions. Salt stress is one of the primary factors that adversely affects plant growth and the photosynthesis process. The two phases of plant photosynthesis are known as the light and carbon reactions. The light reaction in higher plants is mainly caused by the cooperation of four transmembrane multi-subunit protein complexes located in the thylakoid membrane of the chloroplast: PSII, cytochrome b6f, PSI, and ATP synthase [30,31]. The unstable chemical energy (ATP and NADPH) generated via the light reaction is transformed by the carbon reaction into more stable forms that are kept in glucose molecules [32,33]. Changes in these chemical reactions due to salt stress reduce the chlorophyll contents and photosynthesis efficiency of plants [34]. In the present study, the electrolyte leakage was lower and the chlorophyll content was higher in melatonin (MT)-treated plants under salt stress conditions. The chl-a content was found to be lower than the chl-b content throughout the study, which has been confirmed in *Ginkgo biloba* and in sugar beet [35,36], where the same trends for chl-a and chl-b were observed when MT was applied under salt and drought stress. The differences in chl-a and chl-b contents are also associated with the lower illumination of light intensity under stress conditions. The chl-a content was found to be lower than the chl-b content due to a high CO_2_ concentration and less absorption of light by plants in an artificial growth chamber of wheat [37], and this was also confirmed in broccoli plants exposed to low light intensity, which showed minimal chlorophyll contents [38]. The differential content of chlorophyll a and b further underscores the complexity of chlorophyll regulation across plant taxa. This leads to intriguing questions regarding the genetic, physiological, and ecological factors driving such variations. Future investigations could explore the molecular mechanisms underlying chlorophyll biosynthesis, turnover, and pigment–protein interactions in Brassica. Additionally, comparative analyses of chlorophyll metabolism pathways and regulatory networks in these species may elucidate species-specific adaptations to diverse environmental conditions and developmental stages. 

The application of exogenous MT has been reported to show positive effects on reducing electrolyte leakage, enhancing chlorophyll contents, and protecting the chloroplast structure during salt stress in various crops [39]. MT is one of the phytohormones that plays a significant role in the regulation of plant photosynthesis- and chlorophyll-related genes during salt stress, protecting the plant chloroplast structure for normal photosynthesis and improving growth and yield in tomato and maize crops [40,41]. In the present study, the majority of photosynthesis genes were upregulated in the early stages of stress conditions but downregulated after 7 days. MT treatment with salt stress showed the maximum number of genes upregulated throughout the study. The *PsbY* (*BraC02g018500*), *PsaA* (*BraC10g008090*), *PetH* (*BraC09g024280*, *BraC09g039990*), *beta* (*BraC05g022800*), *alpha* (*BraC01g042040*), *gamma* (*BraC02g028860*, *BraC08g030620*, *BraC09g024970*), and *delta* (*BraC07g014400*) genes were upregulated in the MT-treated plants under salt stress when compared with the other treatments, suggesting that these genes had an important role in the plants’ photosynthesis activities. It has previously been reported that the downregulation of these key genes reduced the activities of PSI, PSII, and electron transport in *B. campestris* during high-temperature stress [42].

Exogenous MT stimulated oxidative stress and antioxidant enzyme content to protect the plants during salt stress. The ascorbic acid contents and APX activity were boosted when the plants were exposed to salt stress in combination with MT, and they showed a lower H_2_O_2_ content and less DAB staining after 7 days, suggesting that MT reduced the ROS level to minimize oxidative stress and increase the antioxidant enzyme activities in *B. campestris* during salt stress. Exogenous MT application has been reported to stimulate various physiological and biochemical processes in different crops. MT may protect antioxidant enzymes from oxidative damage by enhancing the efficiency of the mitochondrial electron transport chain, lowering electron leakage, and reducing free radical production [43]. Exogenous MT application’s role has been reported in oxidative stress, redox signaling, ion homeostasis, and enzyme regulation in rice, okra, soybean, cucumber, tea, and maize crops, resulting in reduced oxidative damage under salt stress conditions when compared to control [21,43,44,45,46,47]. Ascorbic acid and APX are the major antioxidants that play a crucial role in resistance accumulation under stress conditions. In the present study, the ascorbic acid content and APX activity were higher when MT was applied, confirming its protective role in salt stress. Ascorbic acid is necessary for a number of biological processes, helps to reduce and regulate the synthesis of reactive oxygen species, and may even support a number of physiological processes in plants, including photosynthesis [48]. AsA production occurs in mitochondria in plant cells via a variety of possible pathways, i.e., the glucose pathway and the ascorbate glutathione pathway [49,50,51,52,53]. In the current study, we identified glucose and ascorbate glutathione pathway key genes at the transcriptional level, which showed their crucial role in ascorbic acid metabolism when MT was applied in salt stress. The D-glucose 6-phosphate (*BraC03g053670*) and UDP-d-glucuronate (*BraC02g029240*, *BraC03g042120*, *BraC07g009210*, *BraC02g029240*, *BraC03g024040*, *BraC01g007210*, and *BraC03g058060*) were the first enzymes that produced L-galactone-1, 4-lactone and oxidized into ascorbic acid, while ascorbate peroxidase contributed to the ascorbate–glutathione cycle via the conversion of H_2_O_2_ to water through releasing ascorbate to monodehydroascorbate (MDHA). MDHA was subsequently changed to ascorbate by the MDHA reductase (*BraC03g001520*, *BraC06g044350*, *BraC09g012760*, and *BraC09g051560*) enzyme and converted into dehydroascorbate reductase (DHAR) (*BraC06g014730*, *BraC08g029030*, and *BraC10g024570*) to oxidize into ascorbic acid, which suggests that MT played a significant role in the ascorbic acid metabolism during salt stress in *B. campestris*. Recent studies have shown that MT reduced the toxicity of cadmium in wheat through the ascorbic acid and glutathione pathways and improved drought tolerance and photosynthesis in maize seedlings [54,55,56,57], which was further confirmed in cucumber, where MT alleviated chilling tolerance by regulating the ascorbate–glutathione cycle and photosynthetic electron flux [58]. APX contributes to the ascorbate–glutathione cycle via the conversion of H_2_O_2_ to water by releasing ascorbate to MDHA, which is then converted into an MDHA reductase, DHAR, and oxidized into ascorbic acid [59].

## 4. Materials and Methods

### 4.1. Plant Material and Growth Conditions

The response of *Brassica campestris* L. ssp. chinensis (Var. Suzhouqing) to melatonin (MT) under salt stress was evaluated at the National Key Laboratory of Crop Genetics and Germplasm Enhancement and Utilization, Nanjing Agricultural University, in October 2023, using uniform and healthy seeds. The seeds were germinated in petri dishes, and the seedlings were then transplanted into pots containing a soil mixture (1:1, vermiculite/humus) and grown in a controlled environment with a temperature of 25 °C, a relative humidity of 60%, a 16/8 h light (250 μmol·m^−2^·s^−1^) cycle, and an 8 h dark photoperiod. At leaf stages 3–4, the plants were divided into four different groups, i.e., A, B, C, and D. Group A plants were kept as a control (CK) with a foliar spray of water, and group B and group D plants were exogenously sprayed with 100 µM of melatonin (MT) for 7 days, according to a previous study [60,61]. Then, after 7 days, group C and group D plants were subjected to stress using 200 mM of NaCl for 7 days. The leaf samples were collected in liquid nitrogen after 1 and 7 days and stored at −80 °C for further analysis.

### 4.2. Determination of Electrolyte Leakage and Chlorophyll Contents and Transmission Electron Microscopy

Electrolyte leakage was calculated according to a previous study [62]. First, 0.2 g of fresh leaves were taken in 10 mL of distilled water and put in a shaker incubator for 2 h at 32 °C. EC1 was calculated with an EC meter for all the samples, which were then placed again at 121 °C for 20 min and cooled at 25 °C to measure the EC2. The following formula was used to measure the electrolyte leakage: $$EL\;(\%)=\frac{\;EC1}{EC2}\times 100$$

For chlorophyll content, 0.2 g of fresh leaf extract in 80% acetone was centrifuged for 10 min at 4 °C. The supernatant was collected, and absorbances were taken for chlorophyll a (chl-a) and chlorophyll b (chl-b) at 663 nm and 645 nm with a multi-detection microplate reader, CYTATION3, BioTek USA (BioTek, Winooski, VT, USA). Total chlorophyll content was measured by adding chl-a and chl-b [63]. For transmission electron microscopy, 1–2 mm leaf pieces in fixative solution (2.5% glutaraldehyde in 0.1 M of phosphate buffer) were taken, and after dehydration, the samples were stained and imaged with a ThermoScientific Tecnai F20 microscope, USA [64]. 

### 4.3. Determination of H_2_O_2_, DAB Staining, Ascorbic Acid Contents, and APX Activity

The H_2_O_2_ content was determined through the standard method [65]. Leaf samples were extracted with 5 mL of 0.1% trichloroacetic acid (TCA) and then centrifuged at 12,000 rpm for 15 min. Then, 0.5 mL of supernatant was combined with 1 mL of 1 M potassium iodide and 0.5 mL of 10 mM phosphate buffer (pH 7.0), and sample readings were taken at 390 nm with a multi-detection microplate reader, CYTATION3, BioTek USA. The DAB staining was carried out by soaking the leaves in DAB buffer (1 mg/mL) for 8 h at 100 rpm in dark conditions, then replacing it with a bleaching solution (ethanol/acetic acid and glycerol = 3:1:1), which was boiled at 95 °C for 15 min to remove the color, and the images were taken [66]. Ascorbic acid content was measured through high-power liquid chromatography (HPLC, Waters, MA, USA) [67]. Leaf tissues, 0.1 g in 1 mL of 0.1% oxalic acid, were precooled in an ice bath and centrifuged at 12,000 rpm for 20 min at 4 °C. The supernatants were collected, filtered through a 0.45 µm syringe membrane filter, and used for HPLC to measure the ascorbic acid content. For APX activity, 0.1 g of leaf sample was grinded in liquid nitrogen; precooled potassium phosphate buffer was added and centrifuged at 13,000 rpm for 15 min at 4 °C; and the enzyme extract was used for APX activity at 290 nm with a multi-detection microplate reader, CYTATION3, BioTek USA [68].

### 4.4. RNA Extraction and cDNA Library Preparation

Total RNA was obtained from the leaves using the RNA kit (Tiangen, Beijing, China) guidelines. For transcriptome sequencing, three biological replicates were used. After that, oligo (dT) was applied to the RNA, and a fragment buffer was mixed in order to synthesize cDNA. After that, the ligation product was amplified using PCR, and pair-end sequencing was carried out using an Illumina sequencing HiSeq 4000 (BGI, Shenzhen, China). The raw data were filtered using the FASTP program (https://github.com/OpenGene/fastp) (accessed on 20 January 2024). The filtering criteria were used to eliminate reads that included adaptors, reads with an N ratio of more than 10%, low mass reads (mass value less than 20), and base proportions higher than 50%. For QC control and further analysis, the Babraham (http://www.bioinformatics.babraham.ac.uk/projects/fastqc) QC program (accessed on 20 January 2024) was used. For the functional enrichment analysis of differential genes, we used an online web tool omicshare (https://www.omicshare.com/tools/) (accessed on 20 January 2024) for DEG, gene ontology, and KEGG pathway enrichment analyses 

### 4.5. Quantitative Real-Time PCR Analysis

Quantitative real-time PCR was performed using TransStart^®^ Top Green qPCR SuperMix, and the acquired cDNA was used as a template for transcriptome analysis. Primers were designed with the online tool genescript (https://www.genscript.com/tools/real-time-pcr-taqman-primer-design-tool) (accessed on 20 January 2024) as mentioned in Appendix A. The q-RT PCR machine CFX96, bio-rad USA (Hercules, CA, USA), was used with three biological replicates. The relative gene expression calculation was carried out according to the 2^−ΔΔCT^ method, as described by [69], and significant differences between treatment and the control were determined by the LSD test, with *p* ≤ 0.05.

### 4.6. Statistical Analysis

The data were analyzed using SPSS 20.0 (SPSS Inc., based in Chicago, IL, USA). Graphs and heat maps were created using SigmaPlot 14.0 and TBtools software v2.085 (https://github.com/CJ-Chen/TBtools) (accessed on 3 February 2024). The measurements were repeated at least three times, independently.

## 5. Conclusions

In this work, we investigated the genetic landscape of biochemical and physiological changes in *Brassica campestris* after exogenous melatonin (MT) application during salt stress using transcriptome profiling. We identified several candidate genes related to photosynthesis and ascorbic acid metabolism that likely regulate the melatonin-induced response to salt stress. Our findings suggest that MT helps as a phytohormone to increase chlorophyll contents, reduce electrolyte leakage, and maintain the chloroplast structure by upregulating key genes involved in the photosynthesis pathway. Similarly, MT also reduces the ROS level and increases ascorbic acid contents and APX activity during salt stress. Additionally, the upregulated expression of genes involved in the ascorbic acid metabolism pathway under MT application during salt stress also highlights important insights into their functional roles in MT-induced salt stress tolerance mechanisms. These findings provide valuable resources for understanding the molecular mechanisms through which melatonin regulates stress-responsive pathways against salt stress, meriting future functional studies in plants.

## Figures and Tables

**Figure 1 ijms-25-05092-f001:**
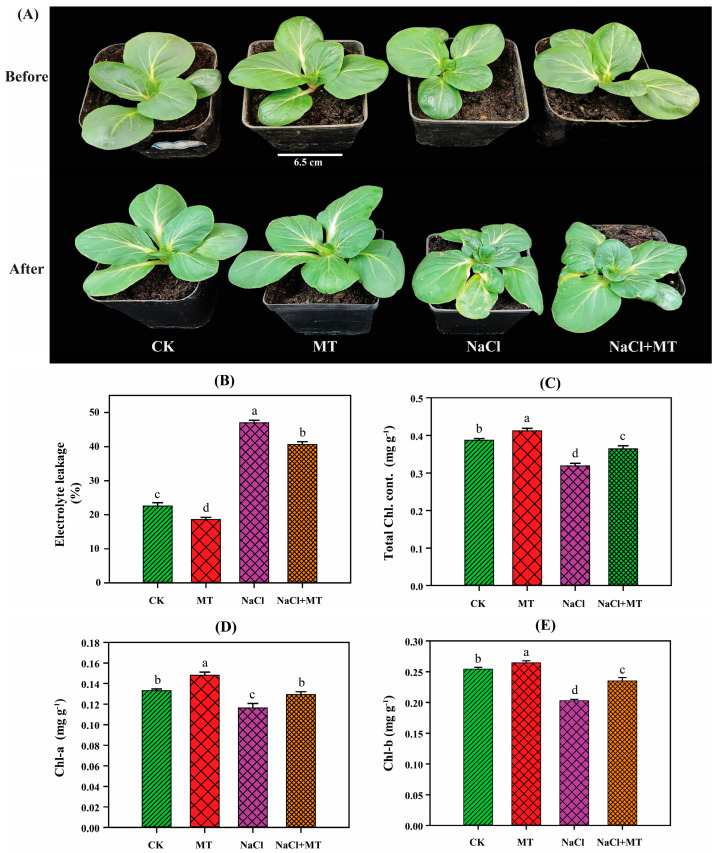
Phenotype, electrolyte leakage, and chlorophyll content analyses in *B. campestris* after 7 days of treatment with exogenously applied melatonin (MT) in the presence and absence of salt stress. (**A**) Physical characteristics (phenotype). Bar: 6.5 cm. (**B**) Electrolyte leakage. (**C**) Total chlorophyll content. (**D**) Chl-a and (**E**) chl-b. The experimental groups were as follows: CK (control), MT (100 µM), NaCl (200 mM), and NaCl + MT (200 mM + 100 µM). Different letters on the bars indicate significant differences between treatment and the control, as determined by the LSD test. The letters a–d indicate the range of significant values from *p* < 0.001 to *p* < 0.05.

**Figure 2 ijms-25-05092-f002:**
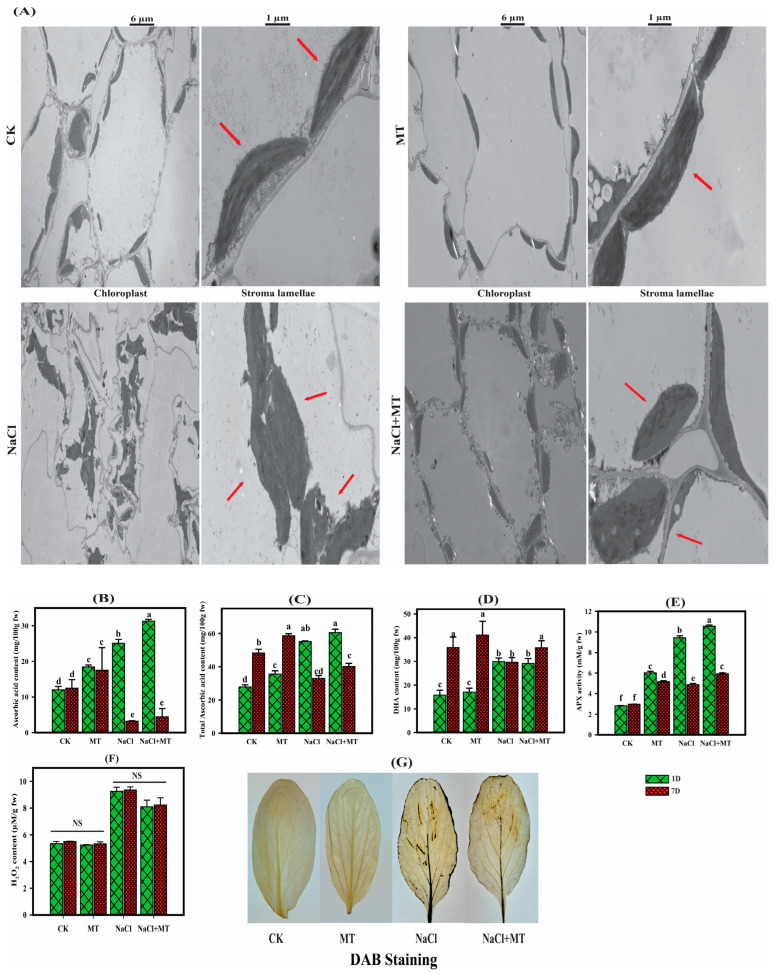
Transmission electron microscopy, ascorbic acid contents, APX activity, and H_2_O_2_ contents after melatonin and salt treatment. (**A**) Transmitted electron microscopy (TEM) at 6 µm and 1 µm. Arrows indicate the cellular structure of chloroplasts with well-defined signs in the leaf of the CK (control), MT (100 µM), NaCl (200 mM), and NaCl + MT (200 mM + 100 µM) treatment groups. (**B**) Ascorbic acid content. (**C**) Total ascorbic acid content. (**D**) DHA content. (**E**) APX activity. (**F**) H_2_O_2_ content and (**G**) DAB staining for H_2_O_2_ in *B. campestris* after MT and salt treatment. The green bars represent 1 day, and the red bars represent 7 days of treatment. Different letters on the bars indicate significant differences between treatment and the control as determined by the LSD test. The letters a–f indicate the range of significant ratios from *p* < 0.001 to *p* < 0.05, and “NS” indicates non-significance.

**Figure 3 ijms-25-05092-f003:**
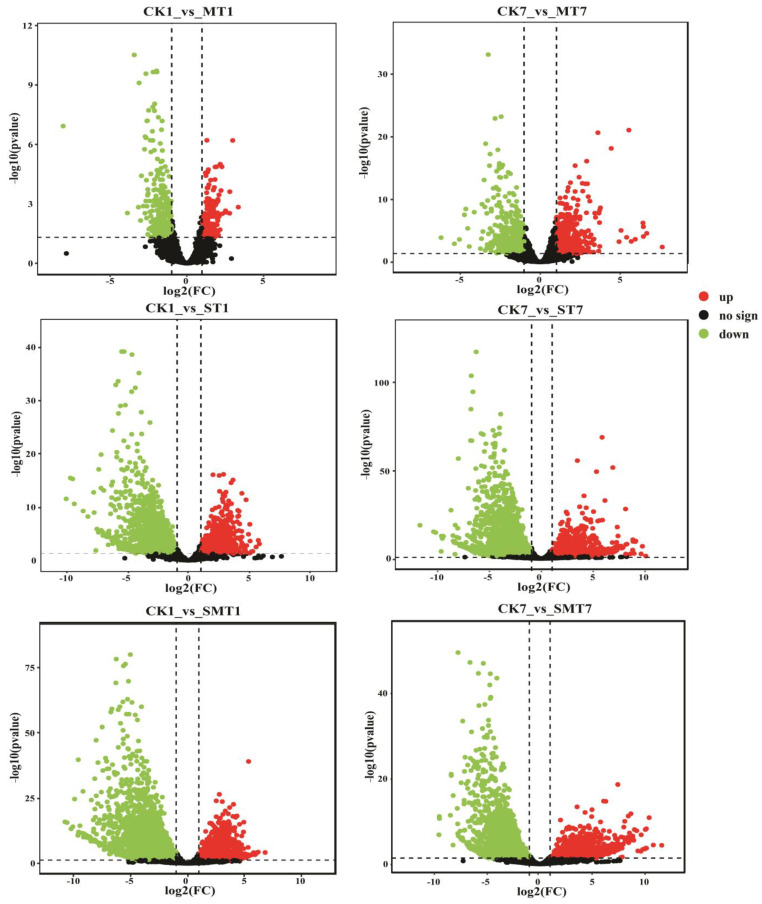
Illustration of a volcano diagram of the differentially expressed genes (DEGs) in the CK (control), MT (melatonin), ST (NaCl), and SMT (NaCl + melatonin) groups of *B. campestris* after 1 and 7 days of treatment. The red color shows upregulated DEGs, while the green color shows downregulated DEGs. The *x*-axis shows the log2-fold change conversion of the values, and the *y*-axis represents the significance value after –log10 conversion. The comparison groups were as follows: CK1_vs_MT1 and CK7_vs_MT7, CK1_vs_ST1 and CK7_vs_ST7, and CK1_vs_SMT1 and CK7_vs_SMT7. The treatments were CK (control), MT (100 µM), NaCl (200 mM), and NaCl + MT (200 mM + 100 µM).

**Figure 4 ijms-25-05092-f004:**
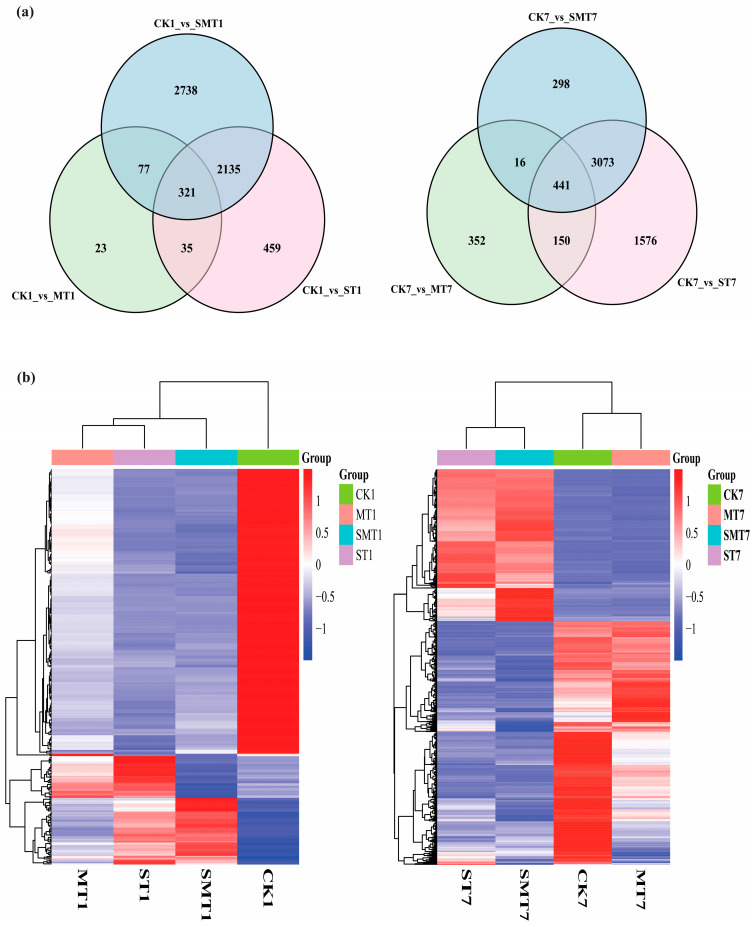
Illustration of (**a**) Venn diagram and (**b**) heat map presenting the number of DEGs expressed in the CK_vs_MT, CK_vs_ST, and CK_vs_SMT groups after 1 and 7 days of treatment. The comparison groups were as follows: CK1_vs_MT1 and CK7_vs_MT7, CK1_vs_ST1 and CK7_vs_ST7, and CK1_vs_SMT1 and CK7_vs_SMT7. The treatments were CK (control), MT (100 µM), NaCl (200 mM), and NaCl + MT (200 mM + 100 µM).

**Figure 5 ijms-25-05092-f005:**
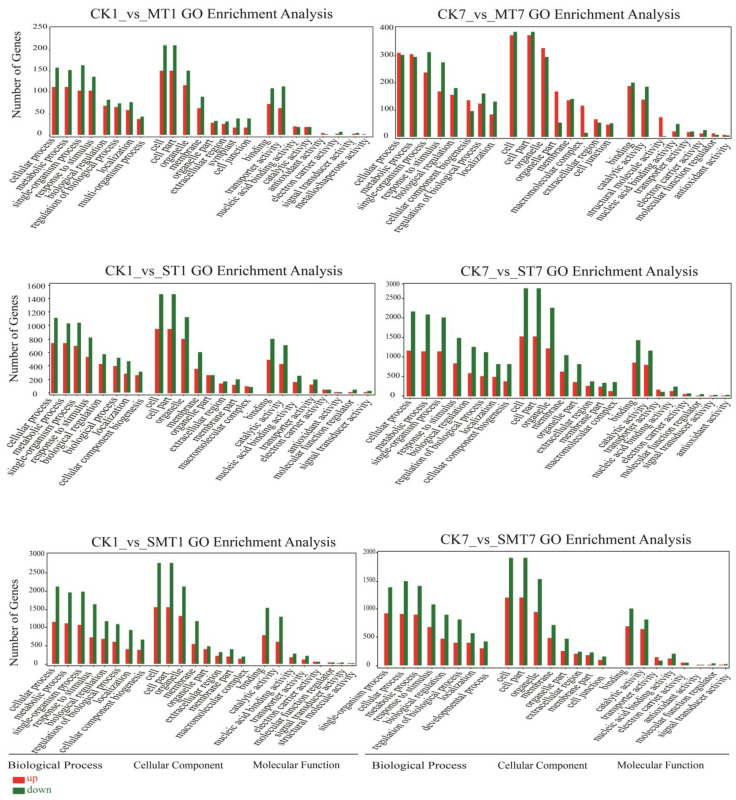
Illustration of GO enrichment analysis for all comparison groups, i.e., CK1_vs_MT1 and CK7_vs_MT7, CK1_vs_ST1 and CK7_vs_ST7, and CK1_vs_SMT1 and CK7_vs_SMT7, after 1 and 7 days of treatment, involving biological processes, cellular components, and molecular functions. The treatments were CK (control), MT (100 µM), NaCl (200 mM), and NaCl + MT (200 mM + 100 µM).

**Figure 6 ijms-25-05092-f006:**
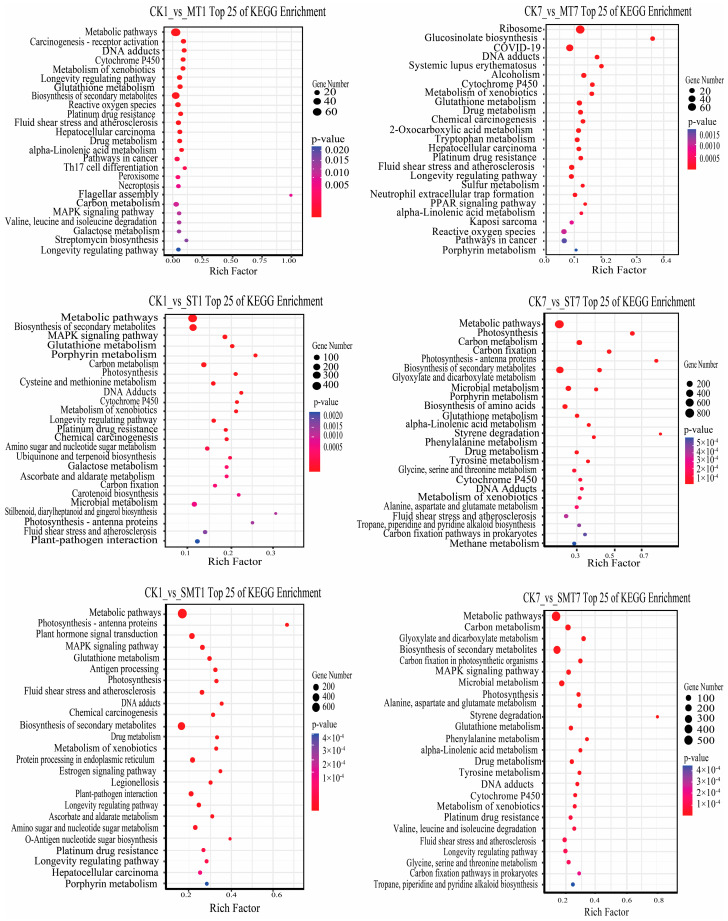
KEGG route bubble diagram showing the top 25 enriched pathways for CK1_vs_MT1 and CK7_vs_MT7, CK1_vs_ST1 and CK7_vs_ST7, and CK1_vs_SMT1 and CK7_vs_SMT7 after 1 and 7 days of MT and salt treatments. The size of the dots indicates the number of DEGs annotated to the KEGG pathway; the axis represents the KEGG pathway; and the abscissa shows the ratio of the number of DEGs annotated to the KEGG pathway to the total number of DEGs. The treatments were CK (control), MT (100 µM), NaCl (200 mM), and NaCl + MT (200 mM + 100 µM).

**Figure 7 ijms-25-05092-f007:**
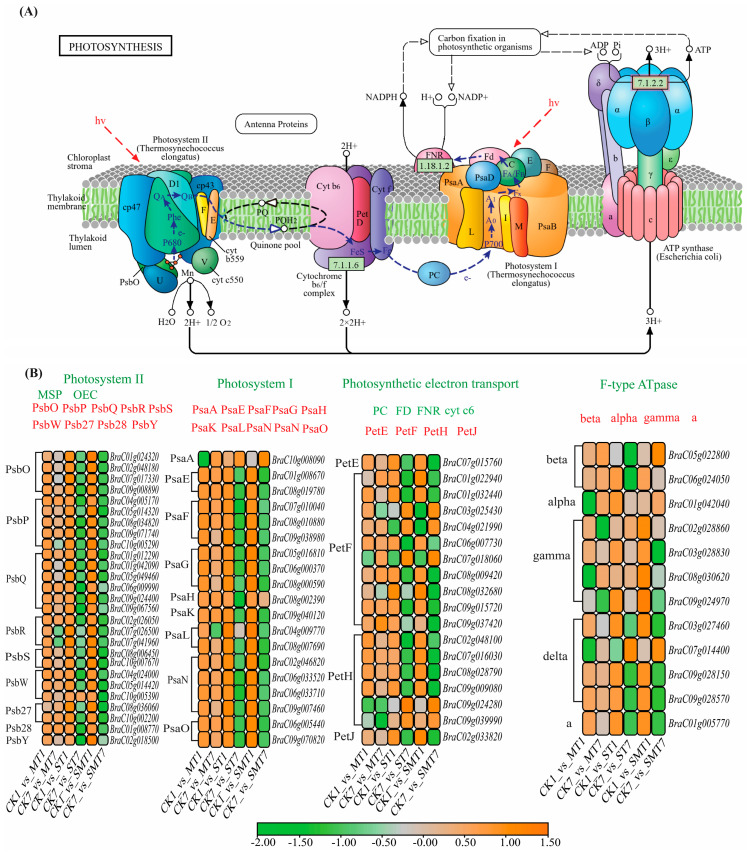
Core photosynthetic pathway and expression of DEGs involved in the plant photosystem. (**A**) Photosynthesis pathway. (**B**) DEGs in photosynthesis. TPM values were changed to percent change values to create this scale. The groups were as follows: CK1_vs_MT1 and CK7_vs_MT7, CK1_vs_ST1 and CK7_vs_ST7, and CK1_vs_SMT1 and CK7_vs_SMT7. The treatments were CK (control), MT (100 µM), NaCl (200 mM), and NaCl + MT (200 mM + 100 µM).

**Figure 8 ijms-25-05092-f008:**
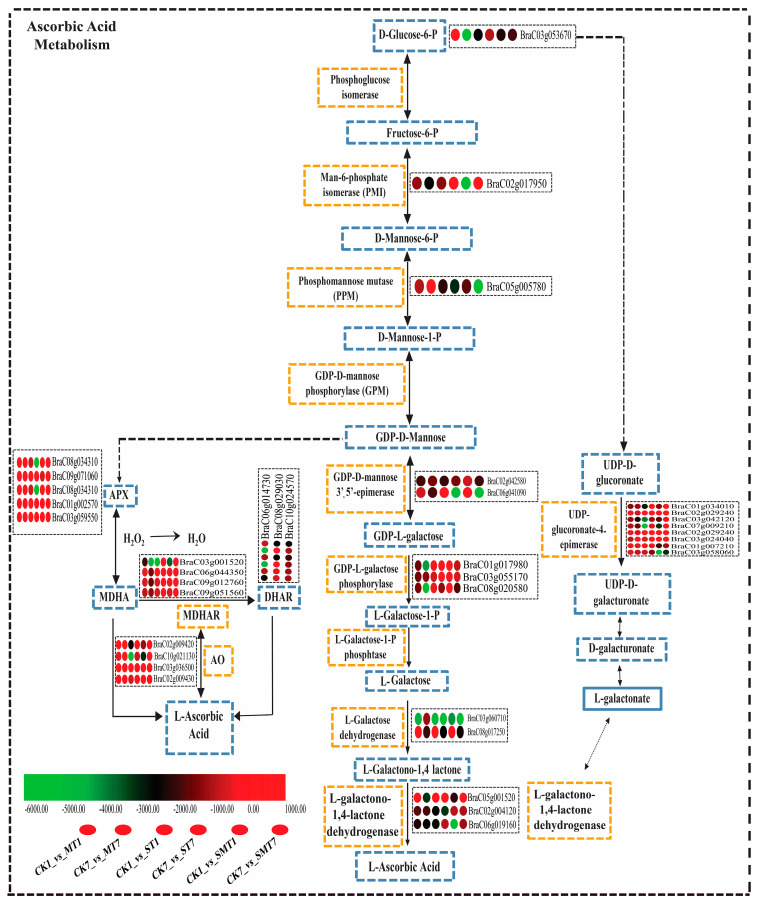
Ascorbic acid metabolism pathway and expression analysis of related DEGS identified after 1 and 7 days of melatonin and salt treatments. DEGs involved in the ascorbic acid metabolism pathway in all comparison groups, i.e., CK1_vs_MT1 and CK7_vs_MT7, CK1_vs_ST1 and CK7_vs_ST7, and CK1_vs_SMT1 and CK7_vs_SMT7. The scale was created on the basis of TPM values converted into a percent change. The treatments were CK (control), MT (100 µM), NaCl (200 mM), and NaCl + MT (200 mM + 100 µM).

**Figure 9 ijms-25-05092-f009:**
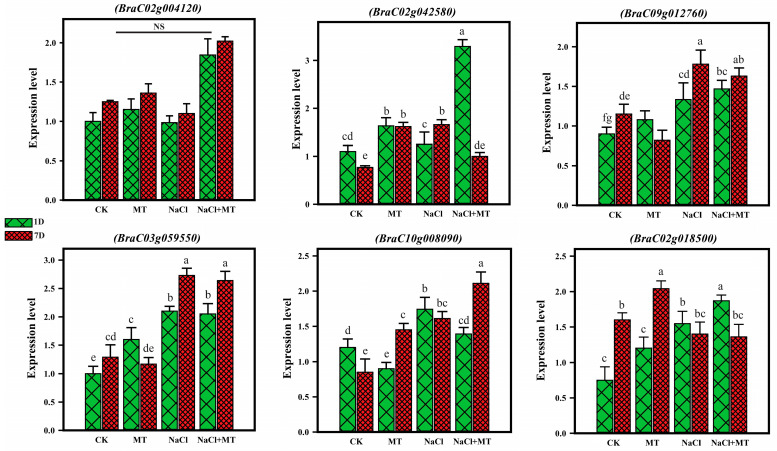
qRT-PCR validation of selected genes under melatonin and salt stress conditions. The treatments were CK (control), MT (100 µM), NaCl (200 mM), and NaCl + MT (200 mM + 100 µM). The green bars represent 1 day, and the red bars represent 7 days of treatment. Different letters on the bars indicate significant differences between treatment and the control, as determined by the LSD test. The letters a–g indicate the range of significant ratios from *p* < 0.001 to *p* < 0.05 and “NS” indicates non-significance.

## Data Availability

The data supporting the results are already mentioned in the main text in the form of figures. The read data for the leaf samples used for the transcriptome analysis and the primer list for qRT-PCR are given in the Appendix A. The RNA-Seq data are available on the NCBI website with the accession number PRJNA1103946.

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
