# Peer review of "Comparative Transcriptome Analysis Reveals the Protective Role of Melatonin during Salt Stress by Regulating the Photosynthesis and Ascorbic Acid Metabolism Pathways in Brassica campestris"

_ijms, 2024, doi:10.3390/ijms25105092_

Round 1

Reviewer 1 Report

Comments and Suggestions for Authors

There is continued interest in your manuscript titled "Comparative Transcriptome Analysis Reveals the Protective Role of Melatonin During Salt Stress by Regulating the Genes Involved in Photosynthesis and Ascorbic Acid Metabolism in Brassica campestris L. ssp. chinensis” which you submitted to the International Journal Molecular Sciences. This study to identify the protective role of melatonin during salt stress in Brassica campestris through its biochemical and transcriptome analysis to provide a new sight for future studies. Your paper has been conditionally accepted. Please carefully revise the following content according to review comments.

1. Abbreviations are written in full the first time they appear in the text, and then followed by a parenthesis in which the abbreviation is written.

2. There should be a space between quantity and unit.

3. Candidate genes screened from comparative transcriptomes are suggested to further validate the function of the candidate genes using experiments such as heterologous overexpression, gene silencing and genetic complementation.

4. Please replace Figure 2A with one of greater clarity.

5. Please replace Figure 4 with one of greater clarity.

6. Please replace Figure 7A with one of greater clarity.

Comments on the Quality of English Language

The English of your manuscript must be improved before resubmission. We strongly suggest that you obtain assistance from a colleague who is well-versed in English or whose native language is English.

Reviewer 2 Report

Comments and Suggestions for Authors

The submitted Manuscript “ Comparative Transcriptome Analysis Reveals the Protective Role of Melatonin During Salt Sress by Regulating the Genes Involved in Photosynthesis and Ascorbic Acid Metabolism in Brassica 3 campestris L. ssp. Chinensis how melatonin alleviates the effects of salinity and suggests the mechanisms behind the phenomenon. Here are the interesting and novel results which could be potentially published. However, the present form of submission can not be recommended for publication without essential changes and major revision or rejection-resubmission.

Main concerns.

Did the Authors apply so called mock treatment with spraying plants by a water solution without melatonin? It would a control of spraying. Please, add and check in any additional experiment.

How did you select the concentration of melatonin for treatment, do you have dose-response curve? Please, provide any information.

Ten times lower concentrations of 10 micromolar melatonin were active in many experiments.

Minor concerns.

1) 4.1. Plant Material and Growth Conditions

a) Please, indicate type of soil/substrate for plants.

b) Please, indicate the level of illumination used and the type of light source.

2) 2.1. Exogenous Melatonin Application Reduces the Electrolyte Leakage and Protected the Photosynthetic Pigments During Salt 72 Stress

Please, use one form of language expression, time: reduced and protected or reduces and protects. The language should be checked in all the text.

3) Figure 1.

Please, add the scale bar for part 1, in cm, meters or miles.

4) Figure 1 and methods.

Electrolyte leakage was calculated according the previous study [47].

Please, describe in full.

5) Figure 1.

Please, change the number of the parts, total chl initially, then chl a and chl b.

6) Figure 1.

Please, check that chl a was lower than chl b. Typically in all the plant ration chl a/ chl b is about 3 that is chl a is 3 times over chl b.

7) Please, indicate everywhere in the text that the leaves of the plants were used only.

Did you do any experiments with roots?

8) Did the Authors measure ion concentrations in the experimental material? It is an essential parameter to measure to characterize the salt treatment.

9) The H2O2 content was determined through the standard method [50].

Please, describe all the methods in full.

10) Methods. Please, add the equipment used with the company which produced etc.

11) Electronic supplementary information: 327

The data supporting the results are already mentioned in the main text and in supplementary files. Transcriptome 328 data can be provided upon the request. 329

There are no supplementary files added to the text.

12) Figure 2.

Please, provide the scale bars for part A, where are 1 and 6 micrometers?

The quality of the figure, part A, is low, the figure legend is distorted.

13) Figure 2. Figure 2: Illustration of transmission electron microscopy, ascorbic acid contents, APX activity and ROS after melatonin 106 and salt treatment.

ROS is not equal to H2O2, please, change the figure legend.

14) Figure 2.

Please, indicate what the red and green bars for in the figure legend. Please, add concentrations of salt and melatonin in the figure legend.

15) Figure 3.

Please, substitute by a better quality figure, it’s not seen.

16) Figure 4.

It’s not readable.

17) All the figure legends. Pls, provide complete description of the experimental treatments similar to figures 1-2.

18) Figure 5.

Not readable.

19) Figures 6-7.

Quality of the figures is very low.

20) Figure 6.

Why do you use the combined figure, part A, from bacterial prokaryotic sources, E.coli, then cyanobacteria Th. Elongates? Pls, try to find more appropriate information for plant species. There could be striking differences between eukaryotes and prokaryotes.

21) Pls, provide the dates when the software programs were accessed.

22) Unfortunately, the Reviewer is not able to recommend the Manuscript for publication and suggests to resubmit it in a while. Plenty of interesting results but the controls are missing and the text is not well presented.

Comments on the Quality of English Language

Readable and understandable but needs improvements.

Reviewer 3 Report

Comments and Suggestions for Authors

The manuscript contains quite interesting research results on the molecular function of melatonin in the regulation of salt stress defence mechanisms. I would like to ask the Authors to suggest future research directions.

Comments

Line 80, please add an explanation of what the letters in the figures mean, correctly mg g-1.

Line 106, 214, please add an explanation of what the letters in the figures mean.

Line 275, in which year was the research carried out? Where? Please give a brief characterisation of the variety tested.

Line 317, P < 0.05.?

References, please remove publications prior to 2014 unless they are necessary to describe the research methodology.

Round 2

Reviewer 2 Report

Comments and Suggestions for Authors

The Authors comprehensively responded to several posed questions of the Reviewer, included the supplementary material and improved the Manuscript but still there are essential points remaining before the Manuscript could be recommended for publication by the present Reviewer.

1) Figure 1. There are 2 (two) parts of the figure marked as Figure 1 C. Please, correct.

2) Quality of figures 3, 5, 6-10 still remains very low. Assuming that the Manuscript has chances to be published, they will not be appropriately reflected at the printed version, still some figure legens are not seen at the highest magnification.

3) The question remains to the journal policy and the Editor. The Authors mention that “Transcriptome 388 data can be provided upon the request.” Most Journals require that the data should be deposited in the www databases prior to the submission or during the submission of the Manuscripts. It doesn’t make sense to discuss the results if they are not available to the readers.

4) Figure 9. The question remains the same while the answer is not sufficient, the impact factor for IJMS is much higher than for Scientia Horticulturae, so the better texts and arguments are needed.

Why do you use the combined figure, part A, from bacterial prokaryotic sources, E.coli, then cyanobacteria Th. Elongates? Pls, try to find more appropriate information for plant species. There could be striking differences between eukaryotes and prokaryotes.

Response: We used combine figure because the genes presented in the heat maps found in the related photosynthesis pathway. Yang et al. (2023) also used the same pathway for blue berries. We followed the same paper for photosynthesis pathway genes analysis.

Yang H, Wei Z, Duan Y, Wu Y, Zhang C, Wu W, Lyu L, Li W. 2023. Transcriptomic and metabolomic investigation of the adaptation mechanisms of blueberries to nitrogen deficiency stress. Scientia Horticulturae. 2023 Nov 1;321:112376..

5) and grown in a 324 controlled environment with a temperature of 25â—¦C, relative humidity of 60% with 16 h light (h (μmol·m−2·s−1) cycle and 325 8 h dark photoperiod.

Please, indicate the illumination of μmol·m−2·s−1 used in numbers, 10, 100, 1000 or 10000? It is important to characterize the plants.

6) Please, discuss the ration of chl a and b in the discussion with the corresponding references. The problem of higher chl b could be linked to the stressed plants due to low illumination level.

7) The text still requires major revisions.

Author Response

The response file is in attachment.

Round 3

Reviewer 2 Report

Comments and Suggestions for Authors

The present Reviewer is well satisfied by the responses of the Authors and the changes done, the main problems were with the ratio of chl a/b and the availability of the transcriptome data, they are solved.

The Reviewer still suggests the Authors to be sure about the low chl a/b ratio, it’s the same high ratio in Ginkgo, e.g. https://www.jkip.kit.edu/botzell/downloads/Pub_Lichtenthaler_2023.pdf but leaves the questions to the Authors and future research.

Language revision is still required, e.g.

These 27 results suggest that exogenous application of MT in plants provide important insight into under-28 standing MT-induced stress-responsive mechanisms…”

Provides, not provide. And so on.

Comments on the Quality of English Language

Language revision is still required, e.g.

“ These 27 results suggest that exogenous application of MT in plants provide important insight into under-28 standing MT-induced stress-responsive mechanisms…”

Provides, not provide. And so on.

Author Response

The response file is attached
